# CD11c is not required by microglia to convey neuroprotection after prion infection

**James A. Carroll** *, James F. Striebel, Chase Baune, Bruce Chesebro, Brent Race

Laboratory of Neurological Infections and Immunity, Rocky Mountain Laboratories, National Institute of Allergy and Infectious Diseases, National Institutes of Health, Hamilton, Montana, United States of America

* carrollja2@niaid.nih.gov

**Data Availability Statement:** All relevant data are within the paper and its Supporting Information files.

**Funding:** This research was supported by the Intramural Research Program of the NIH, National

## Abstract

Prion diseases are caused by the misfolding of a normal host protein that leads to gliosis, neuroinflammation, neurodegeneration, and death. Microglia have been shown to be critical for neuroprotection during prion infection of the central nervous system (CNS), and their presence extends survival in mice. How microglia impart these benefits to the infected host are unknown. Previous transcriptomics and bioinformatics studies suggested that signaling through the heterodimeric integrin receptor CD11c/CD18, expressed by microglia in the brain, might be important to microglial function during prion disease. Herein, we intracerebrally challenged CD11c$^{-/-}$ mice with prion strain RML and compared them to similarly infected C57BL/6 mice as controls. We initially assessed changes in the brain that are associated with disease such as astrogliosis, microgliosis, prion accumulation, and survival. Targeted qRT-PCR arrays were used to determine alterations in transcription in mice in response to prion infection. We demonstrate that expression of *Itgax* (CD11c) and *Itgb2* (CD18) increases in the CNS in correlation with advancing prion infection. Gliosis, neuropathology, prion deposition, and disease progression in prion infected CD11c deficient mice were comparable to infected C57BL/6 mice. Additionally, both CD11c deficient and C57BL/6 prion-infected mouse cohorts had a similar consortium of inflammatory- and phagocytosis-associated genes that increased as disease progressed to clinical stages. Ingenuity Pathway Analysis of upregulated genes in infected C57BL/6 mice suggested numerous cell-surface transmembrane receptors signal through Spleen Tyrosine Kinase, a potential key regulator of phagocytosis and innate immune activation in the prion infected brain. Ultimately, the deletion of CD11c did not influence prion pathogenesis in mice and CD11c signaling is not involved in the neuroprotection provided by microglia, but our analysis identified a conspicuous phagocytosis pathway in the CNS of infected mice that appeared to be activated during prion pathogenesis.

## Introduction

Prions are composed of a host-derived protein (PrP$^C$) that misfolds becoming infectious and transmissible (PrPSc) [1]. Once misfolded, infectious prions recruit additional host PrP$^C$ and

Institute of Allergy and Infectious Diseases. The funders had no role in study design, data collection and analysis, decision to publish, or preparation of the manuscript. There was no additional external funding received for this study.

propagate through a process referred to as seeded polymerization [2, 3]. Prions can infect the lymphatic and nervous systems and are the causative agent for Creutzfeldt-Jakob disease in humans, scrapie in sheep, chronic wasting disease in cervids, and bovine spongiform encephalopathy in cattle. In the central nervous system (CNS), prion disease manifests with deposits of abnormally folded protease-resistant prion protein (PrPres), astrogliosis, microgliosis, neuroinflammation, spongiosis, and neurodegeneration [4–8].

Previously, we demonstrated that microglia were neuroprotective during prion infection in mice and critical to host defense [9]. When microglial numbers are reduced in the CNS using the CSF-1R inhibitor PLX5622, we found that prion-infected mice had reduced survival times by an average of 30 days. Our survival results were later confirmed by Bradford et al. using prion infected Csf1r$^{\Delta FIRE}$ mice [10], which lack brain microglia and resident macrophages in select peripheral tissues (i.e. skin, kidney, heart and peritoneum) [11]. Furthermore, our most recent studies indicate microglia do not play a role in early prion pathogenesis, clearance, or replication but are likely most effective in the later stages of prion disease [7]. Thus, microglia are important to prolonging survival during prion disease, but the molecular mechanism(s) of this neuroprotection is poorly understood.

In a longitudinal study using bulk RNA sequencing of brain samples from C57BL/6 mice infected with prions [12], a STRING clustering analysis of genes with increased expression associated with microglia suggested signaling through the hetero-dimeric integrin receptor CD11c/CD18 (p150,95, αxβ2, complement receptor 4) might be involved. CD11c/CD18 is categorized with the leukocyte β2 integrins, a family of integrins that share the same β-chain (CD18) and can bind multiple ligands such as intercellular adhesion molecule (ICAM)-1 [13], ICAM-4 [14], iC3b [15, 16], fibrinogen [17], collagen [18], and denatured proteins [19].

CD11c/CD18 function has been better characterized in mouse dendritic cells and human/mouse monocytes, and ligand engagement of CD11c/CD18 can lead to phagocytosis [20], respiratory burst [21], and exocytosis of granular content in myeloid cells [22]. Additionally, expression of CD11c/CD18 is also a marker of microglial activation that increases in brain during focal cortical ischemia [23], Alzheimer's disease [24, 25], and neuroinflammation [26]. CD11c, encoded by *Itgax*, is expressed at low levels in surveilling microglia but is increased in microglia associated with amyloid plaques in mouse models of Alzheimer's disease [25]. These CD11c+ amyloid plaque-associated microglia have a gene expression signature that suggests an enhanced capacity for phagocytosis. Moreover, the number of CD11c+ microglia cells increase in brains of mice with experimental autoimmune encephalomyelitis (EAE), a mouse model of multiple sclerosis [27]. The reduction of CD11c+ microglia in the EAE model correlates with an increased rate of disease progression, suggesting that CD11c+ microglia contribute to reducing the process of autoimmune demyelination [28].

Considering this, we postulated that CD11c/CD18 signaling could be an important mechanism in microglial protection during prion infection. Thus, to test this hypothesis we inoculated mice deficient in CD11c (*Itgax*) expression with RML prions and compared them to similarly infected C57BL/6 mice. We monitored gliosis, neuroinflammation, expression of phagocytosis-associated genes, and prion clearance as prion disease progressed. Lastly, we assessed the impact of CD11c deficiency on survival time after intracerebral inoculation of prions.

## Materials and methods

### Mice

All mice were housed at the Rocky Mountain Laboratory (RML) in an AAALAC accredited facility in compliance with guidelines provided by the Guide for the Care and Use of

Laboratory Animals (Institute for Laboratory Animal Research Council). Experimentation followed RML Animal Care and Use Committee approved protocol #2022–003. Mice genetically deficient in expression of CD11c (CD11c$^{-/-}$) were generated at the Baylor College of Medicine and provided by Drs. Wu and Ballantyne, where the mice were backcrossed onto the C57BL/6 strain for at least seven generations [29]. C57BL/6J mice (referred to as C57BL/6) were purchased from Jackson Laboratories.

## Intracerebral inoculations and prion infection

Mice were anesthetized with isoflurane and then injected in the left-brain hemisphere with 30 microliters of RML prion brain homogenates diluted in phosphate buffered balanced saline solution + 2% fetal bovine serum. Following inoculation, mice were monitored for onset of prion disease signs by observers blinded to the experimental groups. Twelve mice per group were designated for survival studies and checked 3–5 times a week prior to clinical signs, then daily after clinical signs appeared. During prion disease, mice typically progress to show clinical signs such as ataxia, tremors, kyphosis, hyperactivity, somnolence, poor grooming, and a reduction in body condition. Mice were euthanized within 30 minutes of observing marked somnolence and/or a body condition score of < 2 (based on a body condition score 5-point scale) [30]. Prion infected mice were listed as USDA Column E in the RML Animal Care and Use Committee approved protocol #2022–003 and euthanized before reaching moribundity. No mice died from prion disease before meeting our criteria for euthanasia. Two mice in the CD11c$^{-/-}$ group designated for survival studies were euthanized early due to dermatitis. Mice reached clinical endpoint ranging from 149 to 162 dpi. At 80 dpi, 120 dpi, and when mice reached the clinical endpoint, they were euthanized and brains removed. Half brain portions were designated for total RNA isolation, placed into 10% neutral buffered formalin (3.7% formaldehyde) for histology, or flash frozen for biochemical analysis.

## RNA isolation and quantitative RT-PCR analysis

Total RNA was isolated from the left hemisphere of mouse brain by dissociation of tissue in 2 ml TRI-reagent (Sigma) following the manufacturer's protocol. Isolated RNA was rinsed in 2 ml 75% ethanol, centrifuged for 10 min at 13,000 x g, and air dried. Total RNA was suspended in 100 ul of DNase reaction buffer (Ambion) and digested with 6 units of DNase I (Ambion) of 30 min at room temperature. RNA was re-isolated and cleaned using RNA Clean & Concentrator-25 column kit (Zymo Research), eluted with 150 μl nuclease-free water with 1 x RNase Inhibitor (SUPERase-In, Ambion), and stored at -80˚C until use.

For quantitative analysis of changes in transcription from brain tissue using qRT-PCR arrays, 400 ng of high-quality RNA from each sample was reverse transcribed to synthesize cDNA using the RT2 First Stand Kit per manufacturer's instructions (Qiagen). Each cDNA reaction was mixed with 2x RT2 SYBR Green Mastermix purchased from Qiagen with RNase-free water to a final volume of 1.3 ml. Ten microliters of the mixture was then added to each well of a 384-well format plate of either the Mouse Inflammatory Cytokine & Receptors super array PAMM-011ZE, Mouse Phagocytosis super array PAMM-173ZE, or Mouse Focal Adhesion super array PAMM-145ZE (Qiagen). These arrays specifically target 84 genes that are involved in inflammation, phagocytosis, or focal adhesins (S1–S3 Data). Each mouse group consisted of an n = 4, with a total of 9,216 individual qRT-PCR reactions performed in these analyses.

qRT-PCR analysis was carried out on an Applied Biosystems ViiA 7 Real-Time PCR System with a 384-well block using the following conditions: 1 cycle at 10 min, 95˚C; 40 cycles at 15 s, 95˚C then 1 min, 60˚C with fluorescence data collection. Melting curves were generated at the

end of the completed run to determine the quality of the reaction products. Raw threshold cycle (Ct) data was collected with a Ct of 35 as the cutoff. Ct data was analyzed using the web-based RT$^2$ Profiler PCR Array Data Analysis from Qiagen. All Ct values were normalized to the average of the Ct values for the housekeeping genes Actb, Gapdh, and Hsp90ab1. Changes in transcription were calculated by the software using the ΔΔCt based method. Statistical analysis was performed using the unpaired Student's t-test to compare the replicate ΔCt values for each gene in the control group versus infected groups. A mean of ≥ or ≤ 2.0-fold change and p-value of ≤ 0.05 were considered significant. For qRT-PCR data, we did not adjust p-values for multiple comparisons since we were interested in only controlling for the individual error rate, where an adjustment for multiple tests is deemed unnecessary.

## Bioinformatic analysis

A Pearson correlation heatmap and average linkage hierarchical cluster analysis were generated using the average ΔCt values for each mouse cohort (n = 4 mice per group) from qRT-PCR results obtained from the Mouse Inflammatory Cytokine & Receptors super array PAMM-011ZE, Mouse Phagocytosis super array PAMM-173ZE, or Mouse Focal Adhesion super array PAMM-145ZE using the web-enabled application Heatmapper [31]. The 75 genes, their fold change, and the p values from the three qRT-PCR arrays (listed above) that were statistically increased in the brains of clinical C57BL/6 mice were uploaded into the web-based bioinformatics application Ingenuity Pathway Analysis (IPA) from Qiagen. Our specific interest with IPA of these upregulated genes was to assess which of the phagocytic pathways were more affected during prion infection in the CNS.

## Immunoblotting and PrPres analysis

Immunoblotting for PrPres was performed at 80 dpi, 120 dpi, and at clinical to compare PrPres levels between prion infected C57BL/6 mice, prion infected CD11c$^{-/-}$ mice, and uninfected control mice. Brain homogenization, proteinase K (50 mg/ml) treatment, gel electrophoresis, protein transfer and immunoblotting with anti-PrP antibody D13 were performed as previously described [9] with the following modifications. In the presented immunoblots the secondary antibody was peroxidase-conjugated sheep anti-human IgG (Sigma) at a 1:10,000 dilution. Gel wells for disease confirmation were loaded with 0.6 mg brain equivalents per lane.

Protein bands were visualized using an enhanced chemiluminescence (ECL) detection system (GE Healthcare) and exposure to film. Band intensities were obtained by taking images of exposed film with a ChemiDocMP Imaging System (BioRad) with a white-light transilluminator plate, and analysis of band intensities was performed using the ImageLab 6.1 software (Bio-Rad). Using ImageLab Analysis Tools, lanes were manually selected, and bands were detected automatically using low sensitivity. Background subtraction was performed using the analysis function and was consistent across all lanes for each gel, and the Lane Profile tool was used to ensure the entirety of each band was represented. Once captured, the numerical data was entered into GraphPad (Prism v9) software to compare the dataset by an unpaired Student's t test.

## Immunohistochemistry

Portions of brain were removed and placed in 10% neutral buffered formalin for 3 to 5 days. Tissues were then processed and embedded in paraffin. Sections were cut using a standard Leica microtome, placed on positively charged glass slides, and air-dried overnight at room temperature. The following day slides were heated in an oven at 60°C for 30 min.

Deparaffinization, antigen retrieval and staining were performed using the Ventana automated Discovery XT stainer. To stain microglia, rabbit anti-Ionized calcium-binding adaptor molecule 1 (IBA-1) was used at a 1:2000 dilution and was a gift from Dr. John Portis, RML, Hamilton, MT. Astrocytes were stained with rabbit anti-glial fibrillary acidic protein (GFAP) (Dako number Z0334) was used at a dilution of 1:3500. Staining for PrP was done using human anti-PrP monoclonal antibody D13 at a dilution of 1:100. Primary antibodies were diluted in PBS containing stabilizing protein and 0.1% Proclin 300 (Ventana Antibody Dilution Buffer). Diluent without antibody and secondary antibody alone were used as a negative controls. Ventana streptavidin-biotin alkaline phosphatase system was used to detect IBA-1 or GFAP with the exceptions that the secondary antibody was goat anti-rabbit Ig, (Biogenex, HK336-9R) and Fast Red chromogen was used. The secondary antibody used to detect anti-PrP antibody D13 was biotinylated goat anti-human IgG at a 1:250 dilution (Jackson ImmunoResearch, West Grove, PA), and streptavidin-biotin peroxidase was used with 3,3-diaminobenzidine (DAB) as chromogen (DAB Map kit; Ventana Medical Systems, Tucson, AZ). Hematoxylin was used as a counterstain for all slides. Slides were examined and photomicrographs were taken and observed using an Olympus BX51 microscope and Microsuite FIVE software.

### Pixel counting and cell counting

IBA-1, GFAP, and PrP stained slides were examined and photographed using Image Scope software. IBA-1, GFAP, and PrP intensity was quantified by ImageScope positive pixel count algorithm (version 9.1). For each stained brain section, a 5-micron thick median sagittal section representing approximately 55 $mm^2$ was evaluated at 1× magnification. Data was reported as the percentage of positive pixels (positive pixels/total pixels × 100) in the section. Comparisons of pixel count data between uninfected and infected cohorts were made by unpaired Student's t test using GraphPad Prism.

## Results

### Integrin CD11c/CD18 expression during prion infection

We first wanted to accurately determine the longitudinal increase in gene expression of the CD11c/CD18 integrin components by qRT-PCR, the gold-standard technology for validating differentially expressed genes after an RNA-seq screen. Our previous results using bulk RNA-seq of brains from prion-infected mice indicated that both *Itgax* (CD11c) and *Itgb2* (CD18) were increased [12], but RNA-seq has several limitations including less dynamic range and greater quantification limits relative to qRT-PCR. Using primers specific to *Itgax* and *Itgb2*, we compared C57BL/6 brain RNA samples from uninfected mice to those infected with prions for 80 dpi, 120 dpi, and at clinical endpoint (Fig 1). The gene encoding CD18 was significantly increased at 120 dpi and at clinical time points, but *Itgax*, encoding CD11c, demonstrated significant increases at all three time points tested. After 80 dpi, *Itgax* expression in the prion-infected brain rose to over 100-fold relative to uninfected mice. These results enhanced our previous findings obtained by bulk RNA-seq and indicated that *Itgax* and *Itgb2* expression was underestimated in our previous analysis.

### Impact of CD11c on prion disease progression

Astrogliosis, microgliosis, and PrPSc deposition are hallmarks of advancing prion disease. To determine if CD11c, which is expressed exclusively by microglia in the brain [32, 33], is involved in microglial neuroprotection during prion infection, we inoculated CD11c$^{-/-}$ mice

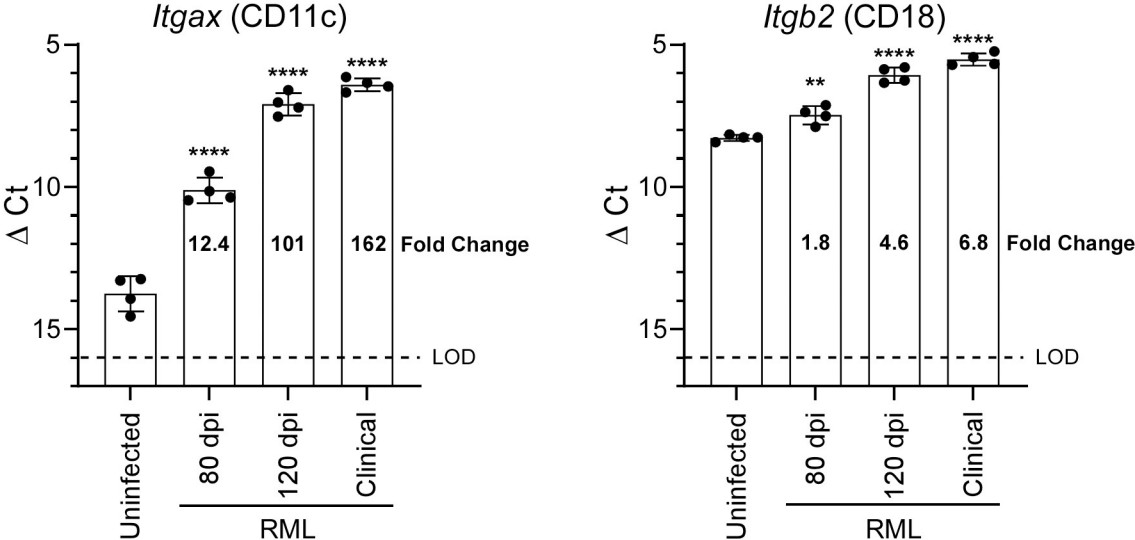

**Fig 1. RNA from brains of C57BL/6 mice infected with prion strain RML at 80 dpi, 120 dpi, and clinical (~150 dpi) was assayed by qRT-PCR to quantify expression of *Itgax* (CD11c) and *Itgb2* (CD18).** The qRT-PCR results are presented as the Delta Cycle Threshold (ΔCt = Ct[target gene]−Ct[reference gene]) values. Each closed circle represents an individual mouse RNA sample (n = 4 for each group). Columns indicate the mean ΔCt and error bars represent one standard deviation. The limit of detection (LOD) for our qRT-PCR assays is represented by the dotted line. The fold change for each mouse group is given relative to uninfected mice. A two-tailed t-test was performed comparing the ΔCt of the uninfected group to the ΔCt of the infected groups, where ** indicates a p value < 0.01 and **** indicates a p value < 0.0001.

with prions and compared them to similarly infected C57BL/6 mice. We stained multiple sagittal sections from infected and uninfected mice with antibodies specific for IBA-1, GFAP, and PrP (Figs 2 and 3). In CD11c-/- and C57BL/6 cohorts staining intensity for IBA-1, GFAP, and PrP deposition increased in accordance with disease progression. Quantitative analysis using positive pixel counts comparing infected CD11c-/- and C57BL/6 mice indicated there was no difference in the detection of IBA-1 (Fig 2B) or GFAP (Fig 3B) at any analogous time point tested. Furthermore, there was no difference in the amount of PrPres as quantified by immunoblot (Fig 4A–4C). Thus, loss of CD11c in microglia does not alter gliosis or prion deposition in the brain as the disease advances.

Reduction or loss of microglia in the brain leads to a quickening of death by ~30 days in prion-infected mice [9, 10]. We assessed if signaling through CD11c was critical to the protection provided by functional microglia. When survival times were compared between cohorts of CD11c-/- and C57BL/6 prion infected mice, we observed no difference in their time to euthanasia due to advanced clinical signs (Fig 4D). Therefore, mice deficient in CD11c were equally susceptible to prion infection as C57BL/6 control mice.

## Evaluation of CD11c loss on transcripts associated with inflammation and phagocytosis during prion infection

Ligation of CD11c in microglia can affect cytokine production [34, 35] and phagocytosis [20], therefore we determined if mice deficient in CD11c altered their gene expression of select targets relative to C57BL/6 mice infected with prions. Using a PCR array panel that profiles the expression of 84 genes involved in inflammation or phagocytosis (S1 and S2 Data), we performed qRT-PCR on RNAs isolated from the brains of CD11c-/- and C57BL/6 mice infected with RML prions for 80 dpi, 120 dpi, and clinical endpoint. These results were compared to RNA samples from the brains of uninfected CD11c-/- and C57BL/6 mice.

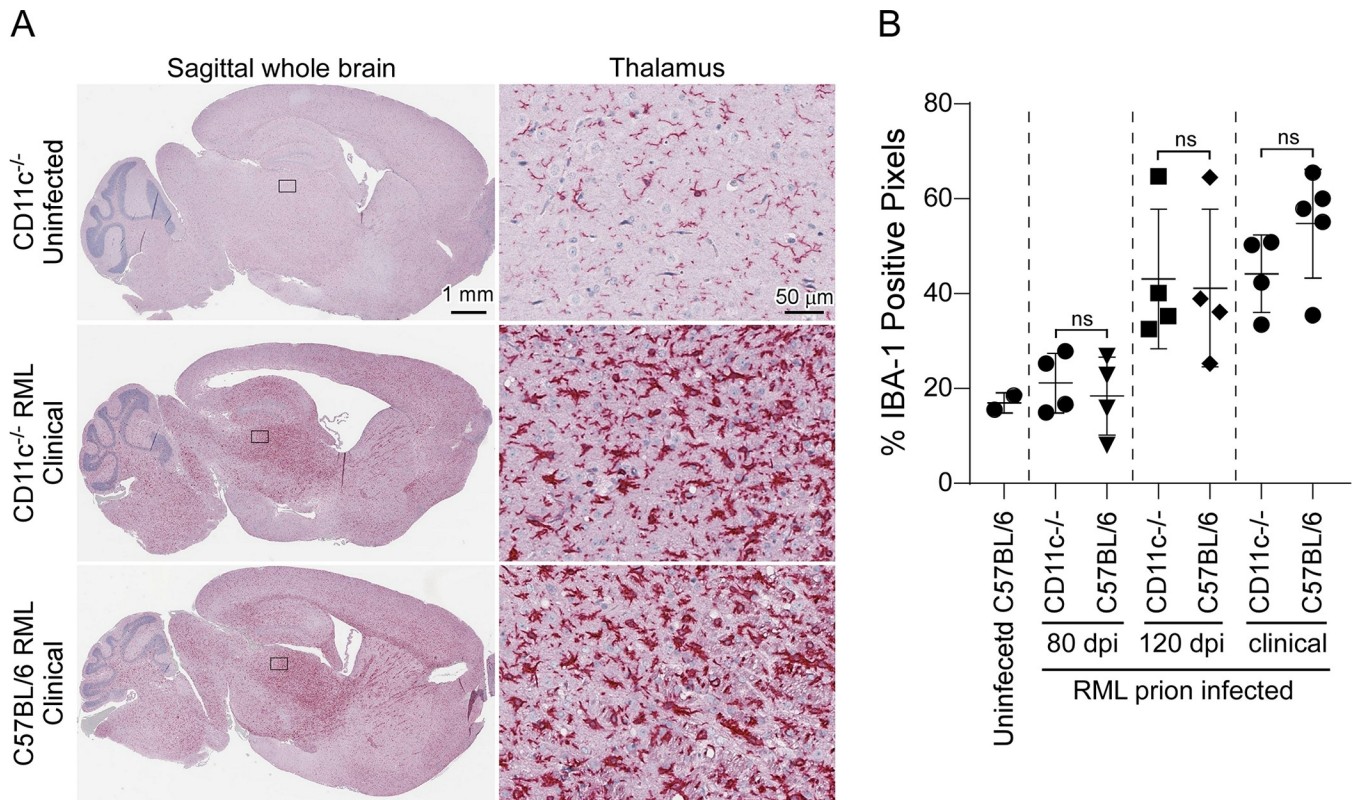

**Fig 2. Immunohistochemical assessment of ionized calcium-binding adaptor molecule 1 (IBA-1) upregulation in the brains of C57BL/6 and CD11c$^{-/-}$ mice during prion infection.** Sagittal brain sections from uninfected mice or mice infected with RML prions at 80 dpi, 120 dpi, and at clinical were stained for IBA-1. Shown in panel A are representative sagittal whole brain sections and thalamus (derived from the rectangle insets from adjacent sagittal sections) from an uninfected CD11c$^{-/-}$ mouse, a clinically infected CD11c$^{-/-}$ mouse, and a clinically infected C57BL/6 mouse. Scales bars are as indicated. Panel B shows the semiquantitative analysis of the percent IBA-1 positive pixels from the entire sagittal section of brain from mice as indicated. Each symbol represents a section analyzed from an individual mouse. A two-tailed t-test was performed comparing the values obtained for CD11c$^{-/-}$ mice to C57BL/6 mice at each timepoint, where ns indicates not significant.

Initially, we were curious if the loss of CD11c altered the baseline expression of the inflammation gene panel in the brain. Comparing uninfected CD11c$^{-/-}$ and C57BL/6 mouse samples, of the 84 genes that constitute the inflammation array only seven genes (*Cx3cl1*, *Cxcr5*, *IL13*, *IL16*, *IL5*, *Lta*, and *Ltb*) were statistically increased in the brains of uninfected CD11c$^{-/-}$ mice (Table 1 and S1 Data). Many of these differences were ≤ 3-fold, but their elevation at baseline did not seem to influence the inflammatory response to prion infection in CD11c$^{-/-}$ mice.

The prion-induced immune response in the CNS of CD11c$^{-/-}$ mice and C57BL/6 mice when compared to uninfected controls demonstrated a comparable increase in inflammatory genes (Fig 5). A summary Venn diagram indicates the significant changes in gene expression of inflammatory mediators in either prion infected C57BL/6 mice (Fig 5A) or CD11c$^{-/-}$ mice (Fig 5B) compared to uninfected control mice. When inflammatory gene expression of CD11c$^{-/-}$ and C57BL/6 mice at 80 dpi and 120 dpi were directly compared (Table 1 and S1 Data), the response to prion infection was similar. At the clinical endpoint of disease, two inflammatory markers (*Ccl8*, and *Cxcl13*) that were increased in all infected cohorts were lower in expression in CD11c$^{-/-}$ mice relative to C57BL/6 mice. As a result, there were a few differences in expression intensity, which can be visualized in the Hierarchical cluster analysis of the inflammation array ΔCt results where mice cluster according to their stage of prion disease. A clade of 38 genes (Fig 5, magenta lines) whose increase in expression correlates with

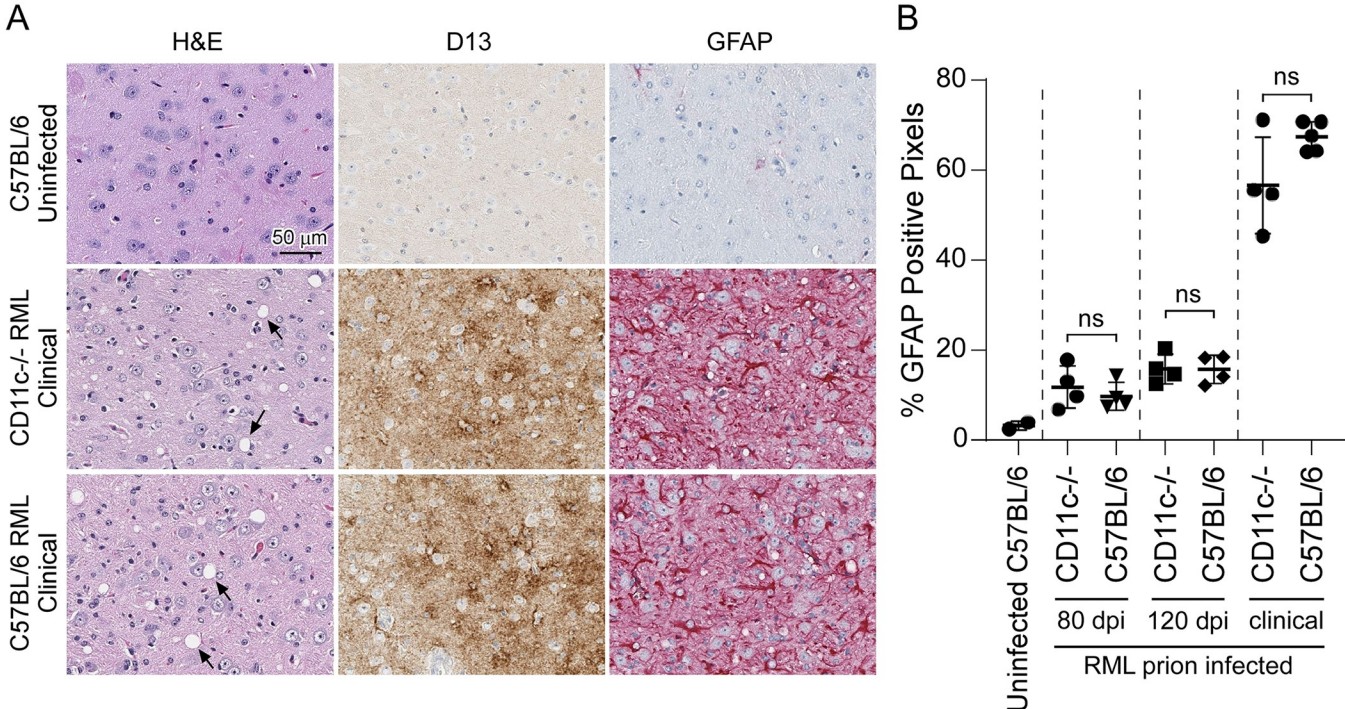

**Fig 3. Histological and immunohistochemical assessment of vacuolation, GFAP upregulation, and prion deposition in the brains of RML infected C57BL/6 and CD11c⁻/⁻ mice.** Brain sections from uninfected mice or mice infected with RML prions at 80 dpi, 120 dpi, and at clinical were stained with hematoxylin and eosin (H&E), antibodies to PrP (D13), or antibodies to glial fibrillary acidic protein (GFAP). Panel A shows representative images from the thalamus of an uninfected C57BL/6 mouse, a clinically infected CD11c⁻/⁻ mouse, and a clinically infected C57BL/6 mouse. The scale bar is as indicated. Arrows indicate vacuoles. Panel B shows the semiquantitative analysis of the percent GFAP positive pixels from the entire brain section from mice as indicated. Each symbol represents an individual mouse section. A two-tailed t-test was performed comparing the values obtained for CD11c⁻/⁻ mice to C57BL/6 mice at each timepoint, where ns indicates not significant.

advancing disease was progressively upregulated in both mouse lines. Ultimately, the overall immune response to prion infection was not considerably altered with the loss of CD11c expression.

CD11c/CD18 is involved in phagocytosis, so a similar qRT-PCR analysis using an 84 gene array panel designed to investigate the expression of genes associated with phagocytosis was utilized. Results indicated that the baseline expression of only *Dock2*, encoding a protein involved with cytoskeletal remodeling required for cellular migration, was decreased 4.8-fold in the CNS of uninfected CD11c⁻/⁻ relative to uninfected C57BL/6 mice. This reduction of *Dock2* in this line of CD11c⁻/⁻ mice has been traced to be a frameshift mutation that arose spontaneously in the Harlan Sprague Laboratories C57BL/6N colony used for backcrossing this mouse line [36]. Thus, changes in *Dock2* were not considered further.

A summary Venn diagram indicates the significant changes in expression of phagocytosis-associated genes in either prion infected C57BL/6 mice (Fig 6A) or CD11c⁻/⁻ mice (Fig 6B) compared to relative uninfected control mice. Most of the phagocytosis associated genes that progressively increased in the CNS during prion disease progression were similar regardless of the expression of CD11c (Fig 6). Only *Il1r1* (an interleukin-1 receptor) was elevated in expression in CD11c⁻/⁻ mice infected with prions at 80 dpi relative to infected C57BL/6 mice (Table 1 and S2 Data). At 120 dpi, no differences in gene expression between CD11c⁻/⁻ and C57BL/6 mice was observed. Though increased overall during prion disease (S2 Data), the genes *Clec7a*, *Itgam*, and *Mcoln3* were significantly decreased in CD11c⁻/⁻ relative to C57BL/6 mice at the

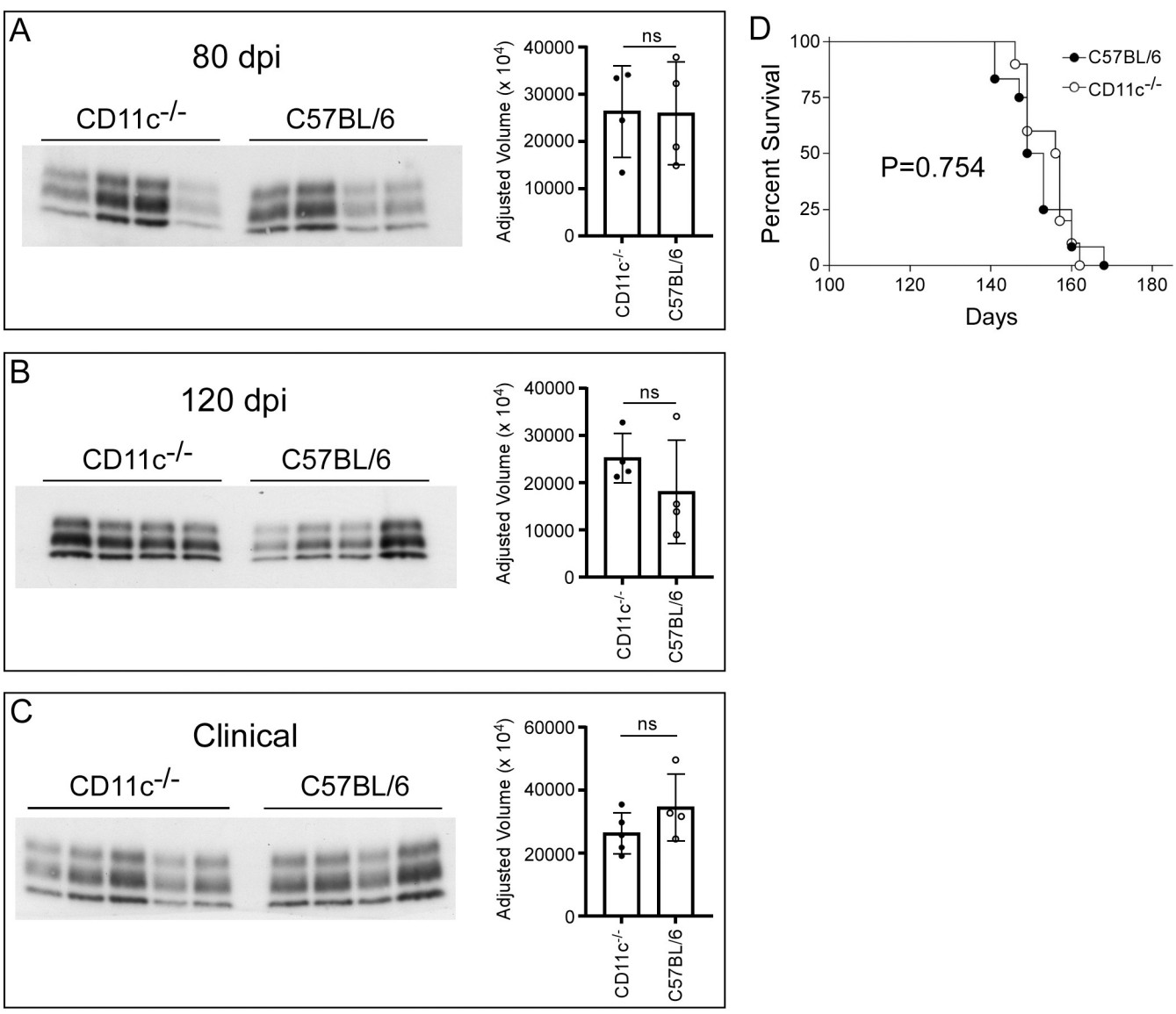

**Fig 4. PrPRes quantification and Survival curves of prion-infected CD11c[-/-] mice compared to C57BL/6 mice.** Brain homogenates from randomly chosen prion-infected mice at 80 dpi (A), 120 dpi (B), and clinical (C) were digested with proteinase K and proteins were separated by SDS-PAGE and immunoblotted using anti-PrP antibody D13. The adjusted volume for the PrPRes bands was measured for each mouse and presented to the right of each immunoblot. A two-tailed unpaired Student's t test was performed comparing the adjusted volumes for CD11c[-/-] mice to C57BL/6 mice at each time point, where ns indicates not significant. Panel D shows the percent survival following infection with 1.0% RML prion brain homogenate. Each survival group included 10 (CD11c[-/-]) or 12 (C57BL/6) mice per group. Statistical analysis (Mantel-Cox, Log-rank test) results are shown within the panel.

clinical stage of prion disease. *Clec7a* encodes a pattern-recognition receptor (Dectin-1) involved in innate immunity [37]. *Itgam* encodes for CD11b, a highly related integrin that is abundantly expressed in microglia and leukocytes and forms a heterodimeric β2 receptor with CD18 (referred to as CD11b/CD18, Mac-1, or CR3) (reviewed in [38, 39]). *Mcoln3* encodes a protein that forms a nonselective ligand-gated cation channel that is likely involved in regulating membrane trafficking events [40, 41]. Hierarchical cluster analysis of the phagocytosis array ΔCt values demonstrates the overall similarity between CD11c[-/-] and C57BL/6 mice in phagocytosis-associated gene expression in response to prion infection, with mouse clusters correlating to their stage of prion infection. A clade of 44 phagocytosis-associated genes (Fig 6,

**Table 1. Summary of genes altered in the CNS of CD11c$^{-/-}$ mice (n = 4) relative to C57BL/6 mice (n = 4) comparing the indicated timepoints during prion infection.**

**Inflammation Array**

| Time point | Genes Altered | Fold Change | P value |
|---|---|---|---|
| Uninfected | Cx3cl1 | 2.0 | $7.9 \times 10^{-5}$ |
| | Cxcr5 | 5.3 | $6.0 \times 10^{-3}$ |
| | Il13 | 2.6 | $1.9 \times 10^{-2}$ |
| | Il16 | 2.0 | $3.4 \times 10^{-2}$ |
| | Il5 | 4.8 | $5.0 \times 10^{-2}$ |
| | Lta | 6.0 | $8.0 \times 10^{-3}$ |
| | Ltb | 3.0 | $8.0 \times 10^{-4}$ |
| 80 dpi | None | - | - |
| 120 dpi | None | - | - |
| Clinical | Ccl8 | -3.4 | $8.2 \times 10^{-3}$ |
| | Cxcl13 | -3.6 | $3.1 \times 10^{-3}$ |

**Phagocytosis Array**

| Time point | Genes Altered | Fold Change | P value |
|---|---|---|---|
| Uninfected | Dock2* | -4.8 | $2.2 \times 10^{-5}$ |
| 80 dpi | Dock2* | -6.7 | $6.9 \times 10^{-5}$ |
| | Il1rl1 | 3.6 | $4.9 \times 10^{-2}$ |
| 120 dpi | Dock2* | -7.8 | $2.5 \times 10^{-5}$ |
| Clinical | Clec7a | -2.0 | $3.3 \times 10^{-2}$ |
| | Dock2* | -13.1 | $3.1 \times 10^{-5}$ |
| | Itgam | -2.7 | $1.1 \times 10^{-3}$ |
| | Mcoln3 | -2.9 | $1.5 \times 10^{-2}$ |

**Focal Adhesion Array**

| Time point | Genes Altered | Fold Change | P value |
|---|---|---|---|
| Uninfected | Itgb6 | 2.1 | $7.5 \times 10^{-3}$ |
| 80 dpi | None | - | - |
| 120 dpi | None | - | - |
| Clinical | Itgam | -2.8 | $3.6 \times 10^{-4}$ |

\* Change in expression in CD11c$^{-/-}$ mice caused by a spontaneous frameshift mutation in Harlan Sprague Laboratories C57BL/6N colony used for backcrossing [35].

magenta lines), many of which are statistically increased regardless of CD11c expression, demonstrated increased expression that correlated with advancing prion infection.

## Assessment of potential compensatory changes in integrin-related gene expression in CD11c$^{-/-}$ mice

Our observation that *Itgam* expression was reduced in prion-infected CD11c$^{-/-}$ mice at the clinical endpoint, lead us to investigate if changes in related integrin genes could be compensating for the deletion of CD11c. We utilized a focal adhesion gene array for qRT-PCR analysis of infected and uninfected C57BL/6 and CD11c$^{-/-}$ mice.

We first determined if there were any genes altered in uninfected CD11c$^{-/-}$ mice relative to uninfected C57BL/6 mice. Only *Itgb6*, encoding Integrin β-6, was significantly altered (increased 2-fold) in the uninfected CD11c$^{-/-}$ CNS (Table 1 and S3 Data). When prion infected CD11c$^{-/-}$ or C57BL/6 mice are compared to their uninfected contemporaries, we detected a limited number of significantly altered genes (Fig 7 and S1 Data). A summary Venn diagram

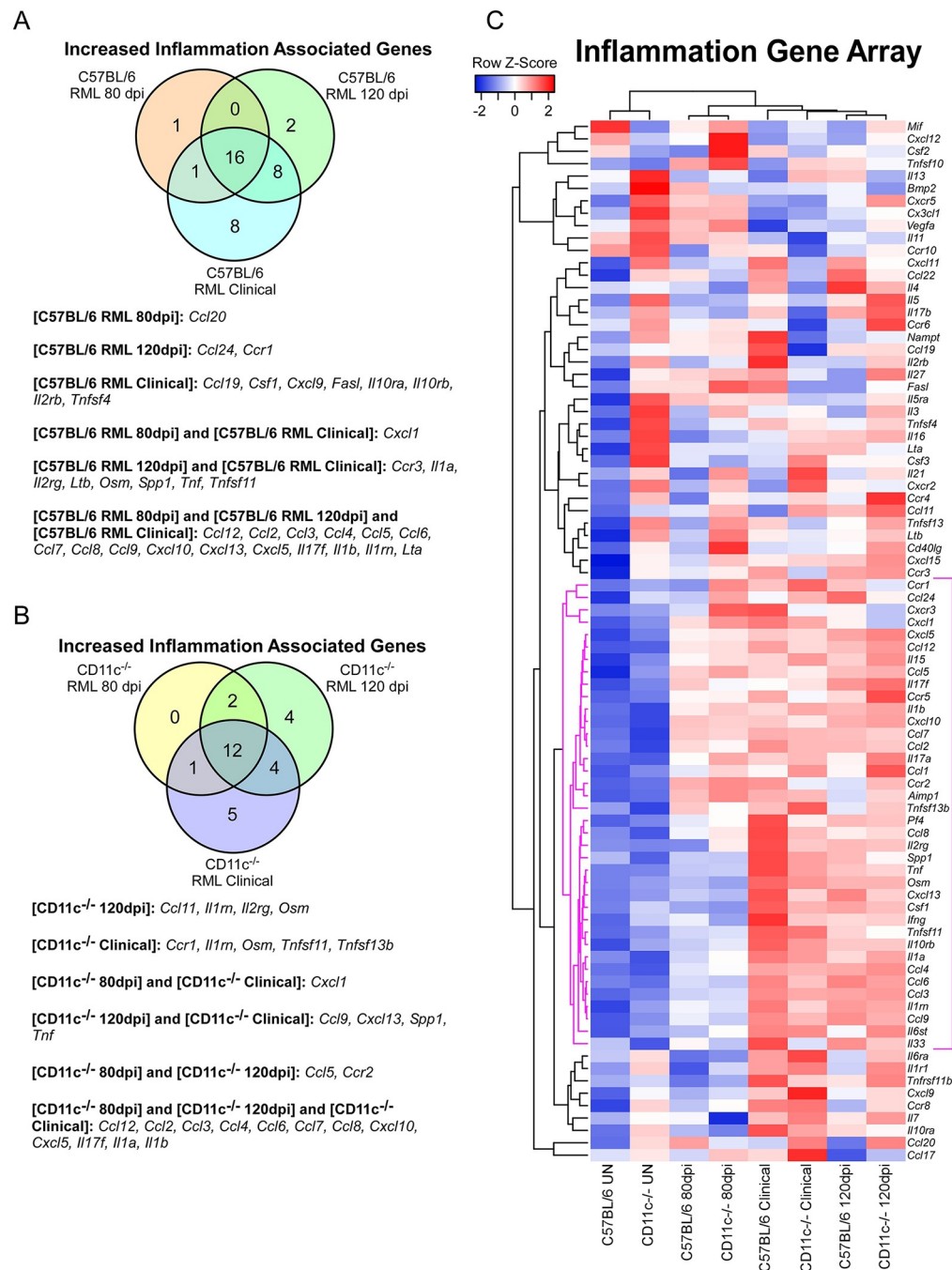

**A**

**Increased Inflammation Associated Genes**

[C57BL/6 RML 80dpi]: *Ccl20*

[C57BL/6 RML 120dpi]: *Ccl24, Ccr1*

[C57BL/6 RML Clinical]: *Ccl19, Csf1, Cxcl9, Fasl, Il10ra, Il10rb, Il2rb, Tnfsf4*

[C57BL/6 RML 80dpi] and [C57BL/6 RML Clinical]: *Cxcl1*

[C57BL/6 RML 120dpi] and [C57BL/6 RML Clinical]: *Ccr3, Il1a, Il2rg, Ltb, Osm, Spp1, Tnf, Tnfsf11*

[C57BL/6 RML 80dpi] and [C57BL/6 RML 120dpi] and [C57BL/6 RML Clinical]: *Ccl12, Ccl2, Ccl3, Ccl4, Ccl5, Ccl6, Ccl7, Ccl8, Ccl9, Cxcl10, Cxcl13, Cxcl5, Il17f, Il1b, Il1rn, Lta*

**B**

**Increased Inflammation Associated Genes**

[CD11c⁻/⁻ 120dpi]: *Ccl11, Il1rn, Il2rg, Osm*

[CD11c⁻/⁻ Clinical]: *Ccr1, Il1rn, Osm, Tnfsf11, Tnfsf13b*

[CD11c⁻/⁻ 80dpi] and [CD11c⁻/⁻ Clinical]: *Cxcl1*

[CD11c⁻/⁻ 120dpi] and [CD11c⁻/⁻ Clinical]: *Ccl9, Cxcl13, Spp1, Tnf*

[CD11c⁻/⁻ 80dpi] and [CD11c⁻/⁻ 120dpi]: *Ccl5, Ccr2*

[CD11c⁻/⁻ 80dpi] and [CD11c⁻/⁻ 120dpi] and [CD11c⁻/⁻ Clinical]: *Ccl12, Ccl2, Ccl3, Ccl4, Ccl6, Ccl7, Ccl8, Cxcl10, Cxcl5, Il17f, Il1a, Il1b*

**C**

**Inflammation Gene Array**

**Fig 5. Alterations in inflammation-associated gene expression in C57BL/6 and CD11c⁻/⁻ during prion infection.** The mouse Inflammatory Receptor and Cytokine qRT-PCR array (Qiagen) was used to monitor CNS gene expression from cohorts of uninfected mice or mice infected with RML prions at 80 dpi, 120 dpi, and at clinical disease. Each cohort consisted of 4 individual mice. Panel A is a Venn diagram showing the genes statistically increased in C57BL/6 mice at 80 dpi, 120 dpi, and at clinical disease relative to uninfected C57BL/6 mice. Panel B is a Venn diagram showing the genes statistically increased in CD11c⁻/⁻ mice at 80 dpi, 120 dpi, and at clinical disease relative to uninfected CD11c⁻/⁻ mice. In Panels A and B the increased genes associated with the sections of the Venn diagram are indicated below. Panel C is a Pearson correlation heatmap and average linkage hierarchical cluster analysis of the row Z-scores obtained from average ΔCt for each cohort. The gene designations are to the right of the heatmap. A core cluster of 38 genes that display a coordinated pattern of expression that correlates with advancing prion disease are indicated with magenta distance lines in the dendrogram to the left of the heatmap.

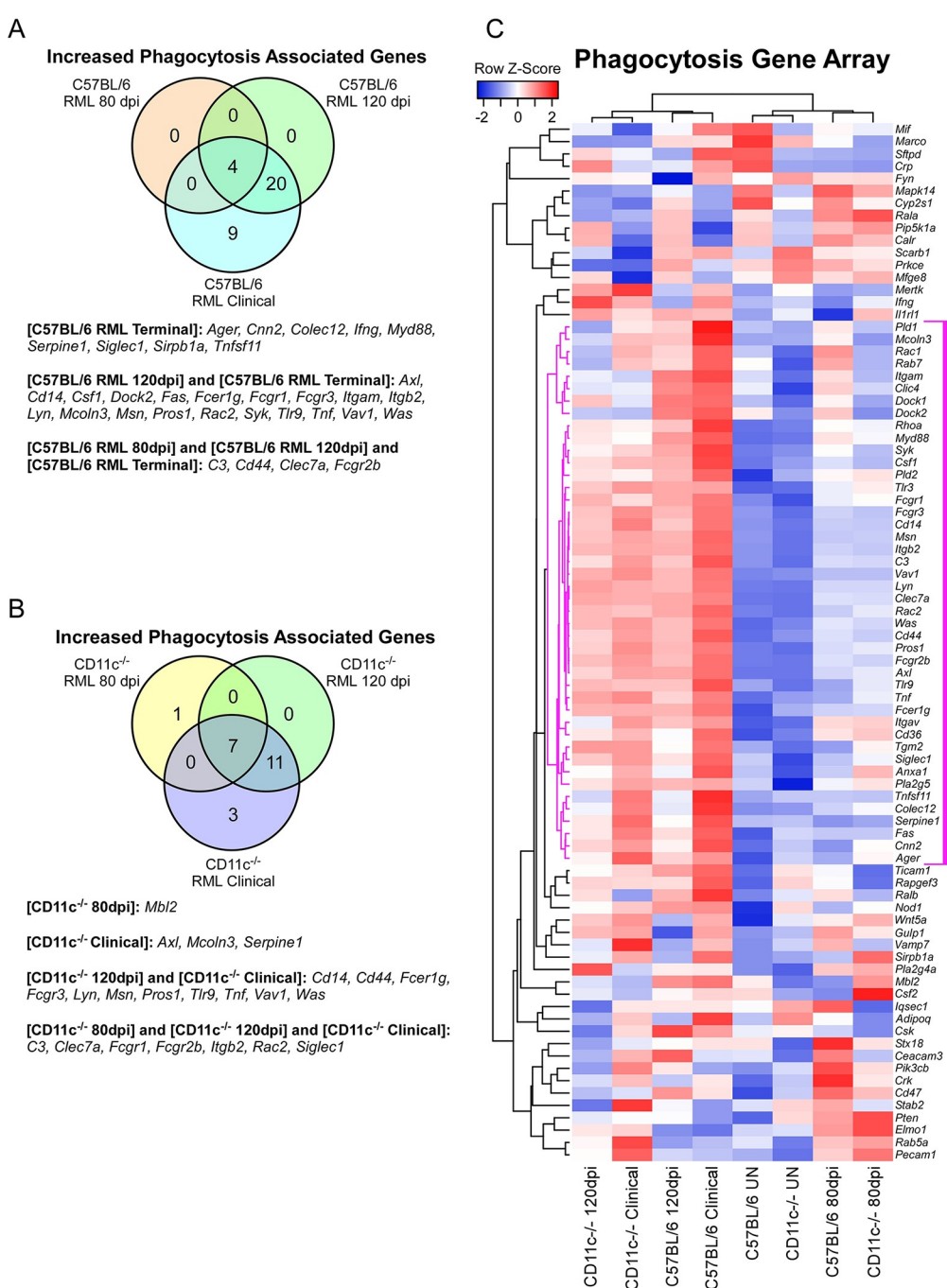

**Fig 6. Alterations in phagocytosis-associated gene expression in C57BL/6 and CD11c<sup>-/-</sup> during prion infection.**
The mouse Phagocytosis qRT-PCR array (Qiagen) was used to monitor CNS gene expression from cohorts of uninfected mice or mice infected with RML prions at 80 dpi, 120 dpi, and at clinical disease. Each cohort consisted of 4 individual mice. Panel A is a Venn diagram showing the genes statistically increased in C57BL/6 mice at 80 dpi, 120 dpi, and at clinical disease relative to uninfected C57BL/6 mice. Panel B is a Venn diagram showing the genes statistically increased in CD11c<sup>-/-</sup> mice at 80 dpi, 120 dpi, and at clinical disease relative to uninfected CD11c<sup>-/-</sup> mice. In Panels A and B the increased genes associated with the sections of the Venn diagram are indicated below. Panel C is a Pearson correlation heatmap and average linkage hierarchical cluster analysis of the row Z-scores obtained from average ΔCt for each cohort. The gene designations are to the right of the heatmap. A core cluster of 44 genes that display a coordinated pattern of expression that correlates with advancing prion disease are indicated with magenta distance lines in the dendrogram to the left of the heatmap.

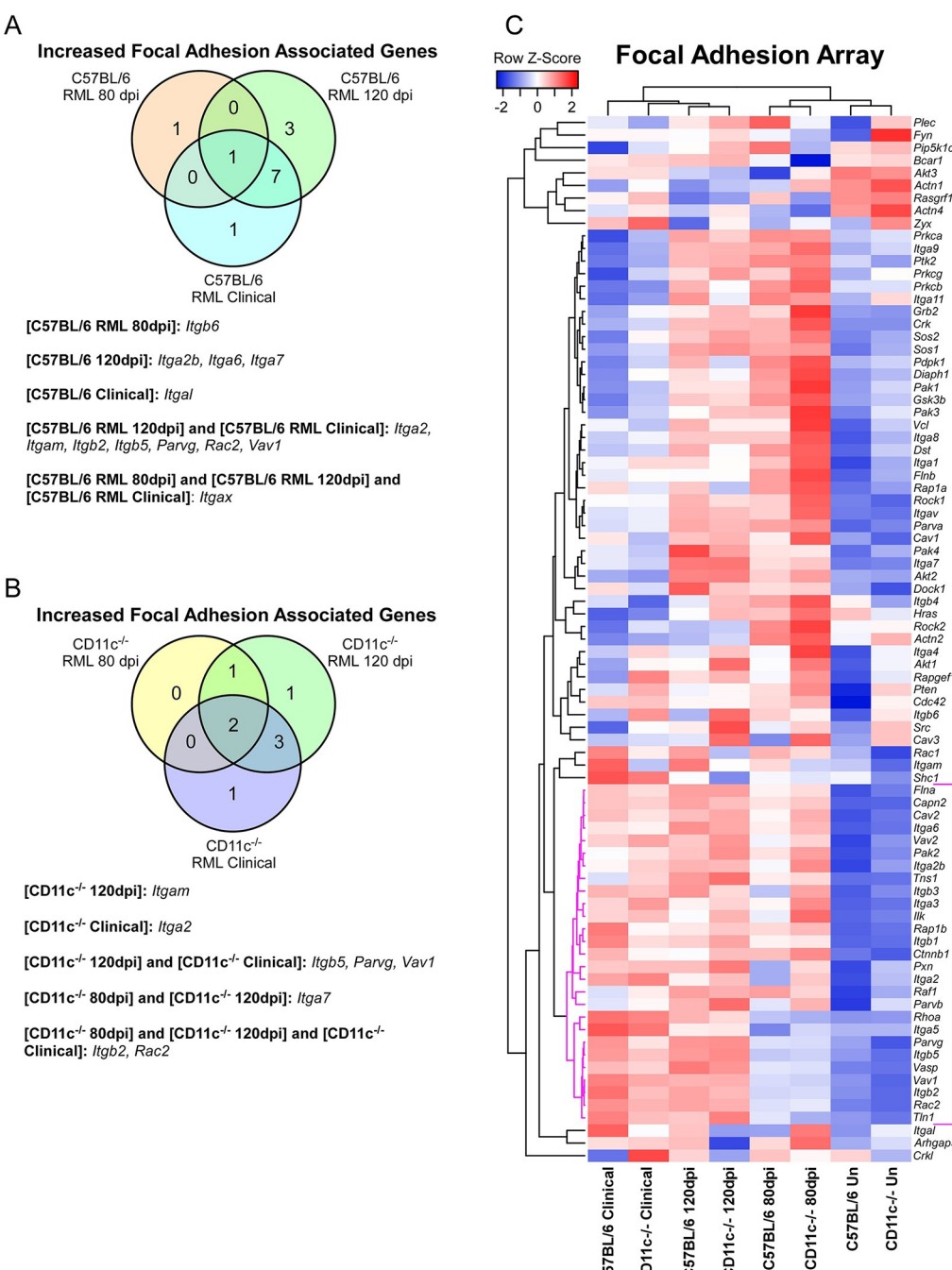

**Fig 7. Alterations in integrin-associated gene expression in C57BL/6 and CD11c$^{-/-}$ during prion infection.** The mouse Focal Adhesion qRT-PCR array (Qiagen) was used to monitor CNS gene expression from cohorts of uninfected mice or mice infected with RML prions at 80 dpi, 120 dpi, and at clinical disease. Each cohort consisted of 4 individual mice. Panel A is a Venn diagram showing the genes statistically increased in C57BL/6 mice at 80 dpi, 120 dpi, and at clinical disease relative to uninfected C57BL/6 mice. Panel B is a Venn diagram showing the genes statistically increased in CD11c$^{-/-}$ mice at 80 dpi, 120 dpi, and at clinical disease relative to uninfected CD11c$^{-/-}$ mice. In Panels A and B the increased genes associated with the sections of the Venn diagram are indicated below. Panel C is a Pearson correlation heatmap and average linkage hierarchical cluster analysis of the row Z-scores obtained from average ΔCt for each cohort. The gene designations are to the right of the heatmap. A core cluster of 27 genes that display a coordinated pattern of expression that correlates with advancing prion disease are indicated with magenta distance lines in the dendrogram to the left of the heatmap.

indicates the few significant changes in expression of integrin associated genes in either prion infected C57BL/6 mice (Fig 7A) or CD11c$^{-/-}$ mice (Fig 7B) compared to uninfected control mice. Hierarchical cluster analysis of the focal adhesion array ΔCt values indicated that mouse group clustering correlated with the degree of prion infection, with a clade of 27 genes that are trending towards increased expression (Fig 7, magenta lines), but only a few of the genes being significantly altered in CD11c$^{-/-}$ or C57BL/6 mice infected with prions.

By directly comparing C57BL/6 and CD11c$^{-/-}$ mice infected for 80 dpi or 120 dpi, there were no statistical differences in gene expression associated with focal adhesions (Table 1 and S3 Data). Lastly, comparing prion-infected mice at clinical endpoint only *Itgam* was determined to be decreased in CD11c$^{-/-}$ mice in agreement with our previous findings using the phagocytosis gene expression array. Thus, using this selected gene array we were unable to identify any major compensatory increases or decreases in integrin related genes in the CD11c$^{-/-}$ mouse line that would confound or explain our results.

### Phagocytosis pathway activated in the prion infected CNS

With similar target genes upregulated in the CD11c$^{-/-}$ and C57BL/6 mice, we compiled the genes that were upregulated in C57BL/6 mice at clinical disease from the three arrays. We analyzed this group of 75 genes using Ingenuity Pathway Analysis software, with an emphasis on phagocytosis, to narrow the field of potential phagocytic pathways that might be involved with prion uptake and/or clearance. The results indicate that numerous phagocytic receptors are upregulated in the prion-infected CNS that feed into Spleen Tyrosine Kinase (SYK) (Fig 8), which mediates signal transduction downstream of cell surface receptor engagement [42, 43]. After surface receptor activation during prion infection, SYK likely phosphorylates the upregulated Vav Guanine Nucleotide Exchange Factor 1 (VAV1), which couples SYK to Rho/Rac GTPases that can initiate actin cytoskeletal rearrangements. This in turn could lead to the activation of the upregulated Ras-Related C3 botulinum toxin substrate 2 (RAC2) and Wiskott-Aldrich syndrome protein actin nucleation promoting factor (WASP, encoded by *Was*). WASP, an effector protein that is important for efficient actin polymerization [44–46], is recruited to the phagocytic cup [47]. Thus, the altered genes we have identified in this study indicate a specific phagocytic pathway that is activated in the CNS in response to prion infection.

### Discussion

Our previous bulk RNA seq analysis indicated that the CD11c/CD18 signaling pathway is activated in the CNS during prion infection [12], and we reasoned that this pathway might contribute to the protective nature of microglia during the later phases of prion disease (post-80 dpi). With our findings using CD11c$^{-/-}$ mice, we would have to reject this hypothesis. Though CD11c$^{-/-}$ is highly expressed in microglia and greatly increased in the brain during infection, it is not essential to convey microglial neuroprotection during prion infection. If it were, we would expect that prion-infected CD11c$^{-/-}$ mice would succumb to disease more quickly as is observed with microglial ablation in vivo [9, 10].

Not only was the survival time unaffected by CD11c deficiency, all measured hallmarks of prion infection in mice were also unaltered. Even with the assessment of transcriptional changes associated with inflammation, phagocytosis, and focal adhesion (all potentially influenced by integrin activation) we observed few changes when comparing CD11c$^{-/-}$ mice to C57BL/6 mice at similar timepoints. Though comparisons of uninfected CD11c$^{-/-}$ and C57BL/ 6 cohorts demonstrated a few altered inflammatory genes, these changes didn't persist after prion infection. Once mice were infected, the examined responses were very similar. Again,

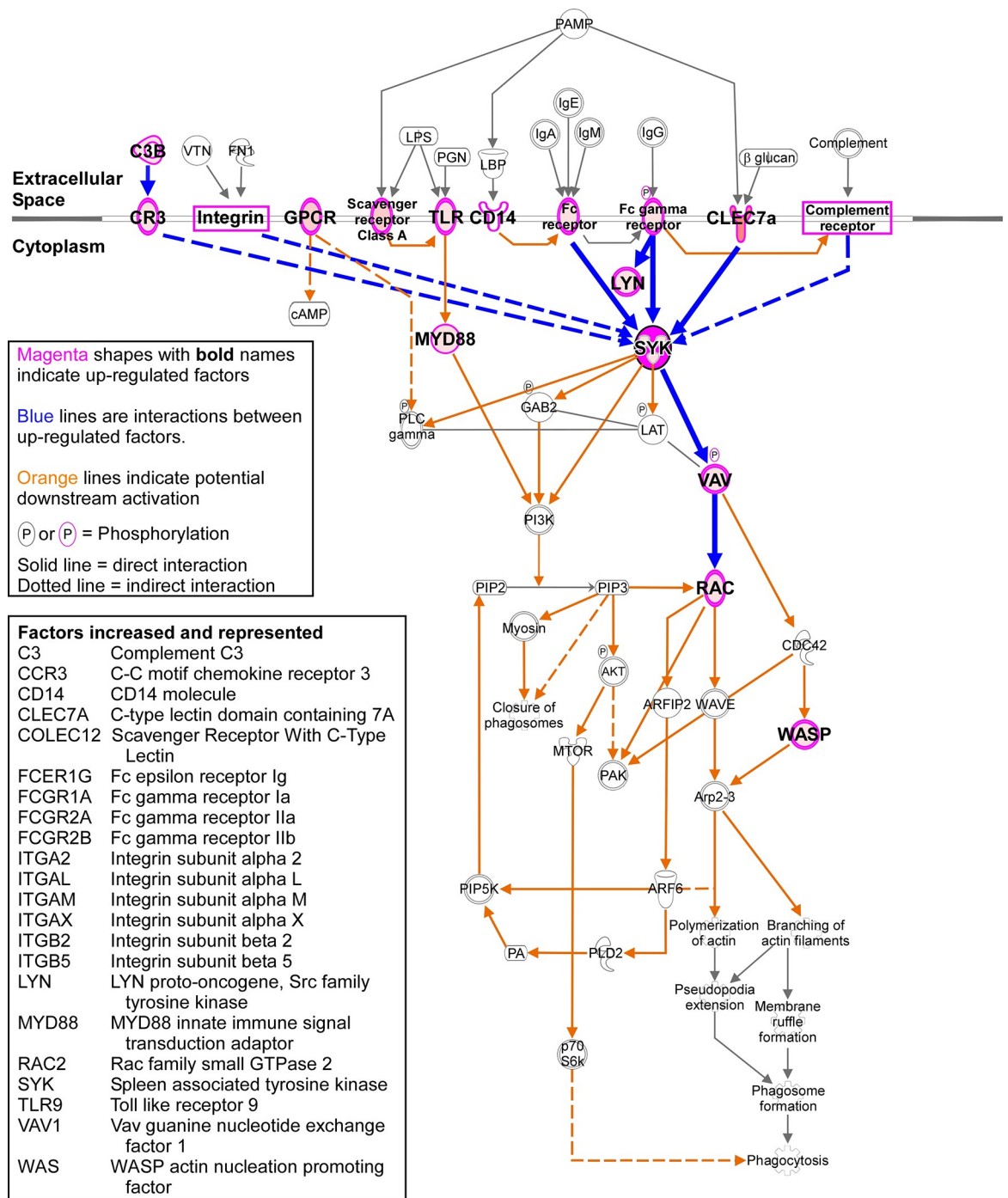

**Fig 8. Phagocytic pathway that is potentially active in the mouse CNS at the clinical timepoint after infection with RML prions.** The 75 genes that were statistically increased in the brains of C57BL/6 mice at the clinical timepoint from the three Qiagen qRT-PCR arrays were interrogated using Ingenuity Pathway Analysis (IPA) web-based software, specifically to evaluate phagocytosis. Aspects of the Phagosome Formation Pathway were statistically enriched (p value = 1.5 x $10^{-17}$) from the analysis, with 22 genes that are featured in the condensed pathway schematic. Magenta shapes represent up-regulated factors, with blue lines indicating direct or indirect interactions between up-regulated factors. The 22 increased genes are listed.

these findings reiterate that the loss of CD11c in mice, and more specifically in microglia of the CNS, does not greatly influence the response to prion infection or the course of disease.

An intriguing aspect of our analysis was the identification of a conspicuous phagocytosis pathway in the CNS of infected mice that appeared to be activated during prion pathogenesis. Phagocytosis is important in development, in tissue homeostasis, during tissue damage, and in response to infection (reviewed in [48, 49]). There are many recognized mechanisms of phagocytosis that are directed by specific ligand-receptor interactions and best studied in professional phagocytes. During prion pathogenesis, multiple receptors that directly or indirectly engage with Spleen Tyrosine Kinase (also increased during infection, [SYK] in Fig 8) were increased in expression. Thus, CD11c might be dispensable in microglial neuroprotection because of overlapping or redundant receptor signaling feeding into the SYK pathway [50].

The tyrosine kinases SYK and LYN, which are increased at the 120 dpi and clinical time points during prion infection, can exert positive or negative effects on multiple cellular processes besides phagocytosis [51, 52]. Both kinases have been implicated in influencing the progression of neurodegenerative diseases such as Parkinson's and Alzheimer's. Often, the effect is dependent on the context and/or the cell type in which the expression/activation is intensified. This is further complicated by the multitude of proposed binding partners and our limited knowledge of their function in the CNS [53].

The LYN kinase, a member of the Src family kinases, is involved in microglia responses and migration induced by neuron-derived α-synuclein in vitro [54]. Furthermore, this migration was later shown to be reliant on Tlr7/8 engagement with α-synuclein leading to Src family kinase activation [55]. Phosphorylated LYN has also been observed in post-mortem brain from Alzheimer's patients [56], and genome-wide association studies identified LYN as an increased risk factor for the development of late-onset Alzheimer's disease [57]. In addition, LYN and SYK are highly enriched in microglia after exposure to amyloid-beta [58, 59]. Activation of these kinases in microglia after toxic amyloid-beta exposure is linked to downstream pathological effects such as oxidative damage, neuroinflammation, and neuronal toxicity (reviewed in [53]). Treatment of APP/PS1 mice with the pan Src family kinase inhibitor AZD0530 attenuated gliosis and synaptic loss [60]. Interestingly, this beneficial effect of AZD0530 treatment was likely due to inadvertent activation of SYK in neurons [61] but Src inhibition in microglia [62], demonstrating diverse effects of AZD0530 in various cell types of the CNS.

In vitro studies by Combs et al. using the synthetic aggregated fragment PrP($_{106-126}$) demonstrated that exposure of post-natal (P3) microglia and THP-1 cells to this toxic peptide aggregate can stimulate LYN/SYK signaling [63]. Furthermore, these studies demonstrated the involvement of the LYN/SYK signaling in the production of a neuronal toxic component produced by THP-1 cells. It is difficult to make comparisons of our work to that of Combs et al. for several reasons: i) we now understand that P3 microglia and immortalized cells are extremely different in their expression signature and responses relative to adult microglia [64], and ii) the biological relevance of this specific controversial PrP peptide fragment not naturally occurring in disease is uncertain [65]. Nonetheless, microglia are now recognized to promote neuroprotection and are beneficial during prion infection [9, 10, 12, 66] and are unlikely to be largely neurotoxic during disease.

Microglia are considered the professional phagocytes in the CNS, but in vitro studies indicate that after prion exposure microglia are impeded in their phagocytic activity [67]. In addition, mouse studies indicate that microglia have limited influence in PrPSc clearance [7, 68]. Collectively, these investigations imply that microglia are incapable of clearing infectious prions from the CNS. Interestingly, the most highly altered myeloid-associated gene was *Clec7a* that encodes Dectin-1, a C-type lectin receptor upregulated in disease associated microglia

that can detect pathogen- and damage-associated molecular patterns [69, 70]. Dectin-1 engagement with endogenous ligands by microglia can dampen neuroinflammation and impart neuroprotection, which differs from responses in macrophages [69–72]. What ligand engages Dectin-1 in the prion-infected brain has not been identified, but Dectin-1 signaling in microglia might contribute to neuroprotection during infection or impact microglial clearance of prions.

If microglia are not influencing prion burden in the CNS, what other cell type in the CNS is capable of phagocytosis and prion uptake? It has been well documented that astrocytes can assume a phagocytic phenotype [73] and express numerous cell-surface phagocytic-associated receptors such as Fc receptors [74], Colec12 [75], Rage [76], and Megf10 [77]. Astrocytes can efficiently internalize PrPSc, even more so than neurons [78, 79], and astrocytes can propagate prions in cell culture [78, 80, 81]. Interestingly, Colec12 that is increased in the clinical prion-infected brain (Fig 6), is a C-type lectin scavenger receptor expressed by astrocytes and microglia postulated to influence amyloid-beta clearance in Alzheimer's disease [75]. Overall, the mechanism of PrPSc uptake by astrocytes is unclear but might involve several of the scavenger receptors identified herein.

A limitation of our study is the use of bulk RNA from mouse brains in our analyses. Because of this approach, we were unable to determine which cell type(s) in the prion-infected brain was responsible for the increase in genes associated with the SYK mediated phagocytic pathway. Nonetheless, signaling through these upregulated scavenger receptors is inadequate to effectively reduce prion propagation in the CNS since the PrPSc burden increases over time and mice eventually succumb to disease. Alternatively, astrocytic uptake of prions using various scavenger receptors might enhance prion propagation and spread, offering a viable site for efficient prion conversion.

## Supporting information

**S1 Data.**
(XLSX)

**S2 Data.**
(XLSX)

**S3 Data.**
(XLSX)

**S1 Raw images.**
(PDF)

## Acknowledgments

Mice deficient in CD11c were originally produced and kindly provided by Drs. Christie Ballantyne and Huaizhu Wu. We thank Jeff Severson for animal husbandry and Tina Thomas for tissue sectioning and immunohistochemistry.

## Author Contributions

**Conceptualization:** James A. Carroll.

**Data curation:** James A. Carroll, Chase Baune, Brent Race.

**Formal analysis:** James A. Carroll, Chase Baune, Brent Race.

**Investigation:** James A. Carroll, Brent Race.

**Methodology:** James A. Carroll.

**Supervision:** James A. Carroll.

**Writing – original draft:** James A. Carroll.

**Writing – review & editing:** James A. Carroll, James F. Striebel, Chase Baune, Bruce Chesebro, Brent Race.

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
