## [Decision Letter · Decision Letter 0]

5 Sep 2023

PONE-D-23-19514CD11c is Not Required by Microglia to Convey Neuroprotection after Prion InfectionPLOS ONE

Dear Dr.Caroll,

Thank you for submitting your manuscript to PLOS ONE. After careful consideration, we feel that it has merit but does not fully meet PLOS ONE’s publication criteria as it currently stands. Therefore, we invite you to submit a revised version of the manuscript that addresses the points raised during the review process.

We look forward to receiving your revised manuscript.

Kind regards,

Human Rezaei

Academic Editor

PLOS ONE

Journal Requirements:

“This research was supported by the Intramural Research Program of the NIH, National Institute of Allergy and Infectious Diseases. The funders had no role in study design, data collection and analysis, decision to publish, or preparation of the manuscript.”

In your cover letter, please note whether your blot/gel image data are in Supporting Information or posted at a public data repository, provide the repository URL if relevant, and provide specific details as to which raw blot/gel images, if any, are not available. Email us at plosone@plos.org if you have any questions

Reviewers' comments:

Reviewer's Responses to Questions

**Comments to the Author**

1. Is the manuscript technically sound, and do the data support the conclusions?

Reviewer #1: Yes

2. Has the statistical analysis been performed appropriately and rigorously? 

Reviewer #1: Yes

3. Have the authors made all data underlying the findings in their manuscript fully available?

Reviewer #1: Yes

4. Is the manuscript presented in an intelligible fashion and written in standard English?

Reviewer #1: Yes

5. Review Comments to the Author

Reviewer #1: The paper by Carroll et al describes prion disease tempo, neuropathogenesis and alterations in brain transcription following experimental inoculation of transgenic mice devoid of CD11c, with the aim of determining whether CD11c-dependant pathways are important in the relative protection offered by reactive microglia. Absence of CD11c was found to be without any notable influence on all features studied. However, the authors identified pathways activated in both wt mice and CD11 ko mice that are of interest, particularly those related to innate immune response and SYK-dependent phagocytosis in the brain.

The results are important as they complement prior work on the role of microglia in prion pathogenesis. The paper is clearly written, technically sound and the conclusions are supported by the data.

One point that would merit further discussion or clarification is the potentially dual role of the activated pathways identified. My understanding is that the authors suggest that the SYK-dependent pathways are potentially “protective” because associated with phagocytosis and potential prion clearance. On the other hand, such inflammatory-related pathways may be detrimental. For example, Combs et al (PMID:9920656) linked neurotoxicity to activation of Lyn/Syk-dependent pathways, following stimulation of primary microglial cell by Ab and PrP fibrils. Both kinases are activated in the present manuscript, raising the question of their real role in prion neuropathogenesis.

What is the genetic background of CD11c ko mice? C57BL/6? This information is important to ensure that comparison of transcriptomes can be made between wt and CD11c ko mice.

6. PLOS authors have the option to publish the peer review history of their article (what does this mean?). If published, this will include your full peer review and any attached files.

Reviewer #1: No

---

## [Author Response · Author response to Decision Letter 0]

19 Sep 2023

Reviewer #1: The paper by Carroll et al describes prion disease tempo, neuropathogenesis and alterations in brain transcription following experimental inoculation of transgenic mice devoid of CD11c, with the aim of determining whether CD11c-dependant pathways are important in the relative protection offered by reactive microglia. Absence of CD11c was found to be without any notable influence on all features studied. However, the authors identified pathways activated in both wt mice and CD11 ko mice that are of interest, particularly those related to innate immune response and SYK-dependent phagocytosis in the brain.

The results are important as they complement prior work on the role of microglia in prion pathogenesis. The paper is clearly written, technically sound and the conclusions are supported by the data.

One point that would merit further discussion or clarification is the potentially dual role of the activated pathways identified. My understanding is that the authors suggest that the SYK-dependent pathways are potentially “protective” because associated with phagocytosis and potential prion clearance. On the other hand, such inflammatory-related pathways may be detrimental. For example, Combs et al (PMID:9920656) linked neurotoxicity to activation of Lyn/Syk-dependent pathways, following stimulation of primary microglial cell by Ab and PrP fibrils. Both kinases are activated in the present manuscript, raising the question of their real role in prion neuropathogenesis.

This is an excellent point by the reviewer. We have expanded the Discussion section by adding 3 paragraphs to cover the involvement of Lyn/Syk in prion and other neurodegenerative diseases. This should help to bridge our findings with RML prion infection of mice and these other relevant neurological disorders. We believe this is a great suggestion by the reviewer and has improved the manuscript. 

What is the genetic background of CD11c ko mice? C57BL/6? This information is important to ensure that comparison of transcriptomes can be made between wt and CD11c ko mice.

We apologize for this oversight on our part. The CD11c -/- mice that we obtained from Drs. Wu and Ballantyne were originally backcrossed onto the C57BL/6 strain for at least seven generations. We have added this important detail in the Materials and Methods section under the subcategory “Mice.”

---

## [Editor Report · Decision Letter 1]

10 Oct 2023

CD11c is Not Required by Microglia to Convey Neuroprotection after Prion Infection

PONE-D-23-19514R1

Dear Dr. Carroll,

We’re pleased to inform you that your manuscript has been judged scientifically suitable for publication and will be formally accepted for publication once it meets all outstanding technical requirements.

Kind regards,

Human Rezaei

Academic Editor

PLOS ONE
---

## [Editor Report · Acceptance letter]

24 Oct 2023

PONE-D-23-19514R1 

CD11c is Not Required by Microglia to Convey Neuroprotection after Prion Infection 

Dear Dr. Carroll:

I'm pleased to inform you that your manuscript has been deemed suitable for publication in PLOS ONE. Congratulations! Your manuscript is now with our production department. 

Kind regards, 

on behalf of

Dr. Human Rezaei 

Academic Editor

PLOS ONE